# The Effect of Maternal Depression on Infant Attachment: A Systematic Review

**DOI:** 10.3390/ijerph17082675

**Published:** 2020-04-14

**Authors:** Andrzej Śliwerski, Karolina Kossakowska, Karolina Jarecka, Julita Świtalska, Eleonora Bielawska-Batorowicz

**Affiliations:** Institute of Psychology, University of Lodz, 91-433 Lodz, Poland; karolina.kossakowska@uni.lodz.pl (K.K.); karolina.jarecka@uni.lodz.pl (K.J.); julita.switalska@uni.lodz.pl (J.Ś.); eleonora.batorowicz@uni.lodz.pl (E.B.-B.)

**Keywords:** maternal depression, prenatal depression, postnatal depression, major depression, infant attachment, systematic review

## Abstract

*Aims and objectives*: The aim of this systematic review was to summarize the key findings of empirical studies assessing the influence of maternal depression on child attachment security measured before 24 months after birth. *Method*: The study followed the Preferred Reporting Items for Systematic Reviews and Meta-Analyses (PRISMA) statement guidelines. A literature search was conducted on the EBSCO (Academic Search Complete; Health Source: Nursing/Academic Edition; MEDLINE; PsycARTICLES) and PubMed databases, with *infant attachment* AND *depression* as search terms with Boolean operators. Study design or sample size did not affect inclusion. After screening, 29 of the 1510 unique publications originally identified were included in the review. *Results*: The studies reveal an equivocal association between maternal depression and child attachment security. Our findings indicate that depression had a significant influence on the attachment style almost only when diagnosed by structured interview: Depression measured by self-descriptive questionnaires was unrelated to attachment style. Furthermore, postpartum depression was found to be significant only when measured up to six months after childbirth. *Conclusion*: The relationship between maternal depression and infant attachment is both complex and dynamic, and the possible negative effects of depression might be compensated by maternal involvement in childcare. Therefore, further studies in this area should employ a reliable methodology for diagnosing depression and a suitable time point for measuring it; they should also adopt a multifactorial and prospective approach. It is important to note that breastfeeding/formula feeding was omitted as a factor in the majority of studies.

## 1. Introduction

Depression is one of the main causes of disability worldwide, and its lifetime prevalence ranges from 20% to 25% in women [1,2]. Women are more vulnerable to depression than men, and the reproductive years are a particularly critical period for its onset [3]. Depression that occurs during pregnancy is referred to as antenatal or prenatal depression, and depression occurring after childbirth is described as postpartum depression (PPD). The prevalence of prenatal depression increases from 5.4% in early pregnancy to 10.0% in late pregnancy [4] and prenatal depression is one of the greatest risk factors for developing PPD [5].

PPD is a common and serious mental health problem, and one that is a source of suffering for both the mother and her offspring. It is often defined as an episode of a major or sometimes minor depressive disorder that occurs anytime within the first year postpartum [6]; however, most episodes begin within two to three months after giving birth [5,7]. The prevalence of PPD has been estimated to range from 9% to 19% depending on the recognition criteria, the period of time under consideration and population type [8]. The prevalence of PPD is 9.6% among women living in high-income countries [9,10] and 19.6% in low-income and middle-income countries [11]. For comparison, the prevalence of MD among non-pregnant women of childbearing age is 4.8% [12]

The majority of studies examining depression suggest that it may be associated with problems in the formation of the bond between mother and child [13,14,15,16], particularly in women with a dual/disorganized attachment style [17]. Depressive mothers develop a less-intense relationship with their children, experience more stress, perceive their children in a more negative way and may assess them as less securely attached than non-depressive mothers [14]. Some experience lowered maternal instinct, as well as greater hostility and aggressive impulses, and a feeling of rejection toward their own children [18]. 

Maternal depression also influences the affective state of the infant and impairs their capacity for repairing states of miscoordination. As a result, the infants develop negative affective states that disrupt their social relations. This impact is associated primarily with severe maternal depression but can also manifest in mothers who have only high levels of depressive symptoms [19]. However, findings from longitudinal studies indicate that the severity and chronicity of maternal depression are associated with higher levels of later childhood behavioural problems [20,21]. Moreover, male infants are more vulnerable than female infants to maternal depression [19,22]. At nine months, the infants of depressed mothers manifest lower social engagement, fewer mature regulatory behaviours and more negative emotionality, and higher cortisol reactivity [23,24]. Some studies have found that the detrimental effect of maternal depression is intensified in mothers with comorbid diagnosis such as anxiety or a personality disorder [25,26,27], and others show that the influence of maternal depression on child development depends on maternal sensitivity and quality of parenting [23,28]. As sensitive parenting requires an accurate and empathic response to signals from the child, the process can be disrupted by changes in the mental state of the parent. 

Some studies have indicated that maternal depression has adverse effects on mother–child bonding and child development, and these examine a variety of factors; however, there is an urgent need to determine the influence of maternal depression on the development of the attachment security of the child and to identify the factors that play crucial roles in this process. Studies in this area tend to focus on individual sets of variables and lack the global analysis needed to gain a comprehensive understanding of the effects of maternal depression. No such systematic review has yet been carried out in this area; existing studies focus either on a broader developmental context [29] or on the effects of depression treatment [30].

The aim of this review is to identify and summarize the key findings of empirical studies assessing the influence of maternal depression on the relationship between mother and child, and on child attachment security, measured up to 24 months after birth. It was decided to use 24 months as a cut-off age for three reasons: (1) to concentrate the analysis on the period of life when children depend mostly on their parents, and thus might be highly affected by any type of parental disturbances or abnormalities; (2) to indicate that intense reorganization takes place in the brain, and the changes are directly related to the attachment style [31]; and (3) to look at the effects of maternal depression in fairly similar age groups and thus control for a child’s developmental stage. Thus, the review places particular attention on the period when depression occurs and its most visible effects for attachment security. It also attempts to identify both the protective and adverse factors that might modify the influence of maternal depression.

## 2. Materials and Methods 

A systematic review was performed of scientific papers examining the influence of maternal depression on child attachment security. Within this review, the term *depression* was applied to either major depression, prenatal depression occurring during pregnancy, or postpartum depression occurring during the first year after delivery. 

### 2.1. Inclusion and Exclusion Criteria

To qualify for review, the studies were required to meet the following three inclusion criteria: (a) the study was empirical and published in an English-language peer reviewed journal from 1 January 1981 to 31 December 2018; (b) child attachment was assessed up to 24 months after birth; (c) the study assessed both maternal depression and child attachment style. Studies were included regardless of the study design or sample size. 

Articles were excluded if they were (a) non-empirical (such as reviews, commentaries, or letters to editors); (b) the child was older than 24 months when measuring attachment style; (c) the study assessed only mother-child bond; (d) the study assessed the efficacy of therapeutic intervention; (e) depression was not considered as a separate variable i.e., depression was aggregated into cumulative risk score, see: [32].

### 2.2. Research Strategy and Data Extraction

The present review followed the guidelines given in the Preferred Reporting Items for Systematic Reviews and Meta-Analyses (PRISMA) Statement [33], using a three-step procedure to identify relevant studies. A systematic search of literature was conducted on the following electronic databases: EBSCO (Academic Search Complete; Health Source: Nursing/Academic Edition; MEDLINE; PsycARTICLES) and PubMed. The search strategy comprised the terms *infant attachment* and *depression* combined with the Boolean operator AND. The search area included the title, abstract, key words, topic and main text of the article (if available). Results were limited to studies on humans.

The first screening of EBSCO and PubMed identified 1718 articles (1102 and 616 hits respectively). After removing duplicates, 1510 unique publications were identified. The abstracts were screened by four reviewers (A.Ś., K.K., K.J., J.Ś.) to obtain potentially relevant publications. Abstracts that did not address the association between maternal depression and child attachment style were excluded. When it was unclear from the titles or abstracts whether all inclusion criteria were met, the papers were subjected to closer inspection: the full texts of these articles were examined before the decision was made to include them or not. Of these works, 68 papers were identified as potentially relevant articles. In this stage of the screening, the articles with abstracts that appeared relevant were selected for full-text evaluation. 

Thirty-nine papers did not meet all inclusion criteria and so were excluded. Twenty-nine papers were, therefore, included in the present review. The following data were extracted to Microsoft Excel spreadsheets with the following specified headings: country where the study was performed, study design, main objectives, information about participants, measurement of attachment style and maternal depression and main results of the research. Another reviewer (EB) screened all data independently to confirm whether the articles met the inclusion criteria. Uncertainty was resolved by consensus among all five reviewers. The study selection process is shown in the flow diagram (Figure 1) based on the criteria of Vu-Ngoc et al. [34]. One of the included studies employed intervention [35]; however, it was included in the present analysis as the measurement of depression and attachment was performed before the intervention.

### 2.3. Risk of Bias Assessment

During the study extraction process, information was collected to determine the risk of bias within each included study and the risk of bias across studies, as well as whether the data obtained from the included studies was valid. The quality assessment influenced the analysis, interpretation and conclusions of this report. 

Multiple publications deriving from the same study were identified using the criteria given by Higgins and Green [36]. Such duplicate publications were detected by the following steps: checking author names, study location and setting, numbers of participants, baseline data and duration of the study. None of the studies included in this review were found to be based on the same sample. 

To assess the risk of bias in individual studies, the final quality assessment included several factors. Firstly, classification to the group was examined: random allocation to groups was not permitted due to the specificity of the research on depression. However, the baseline characteristics of the comparison groups were reviewed, as were the exclusion criteria and attrition rate, existence of blind attachment assessment and the experience of the coders in scoring the attachment style. 

### 2.4. Quality Assessment

Each paper included in the systematic review was assessed according to the quality of the study design, using the criteria outlined in Higgins, Altman and Stern [37]. The review included papers of all study designs, as restricting the range to randomised control trials (RCT) would be unrealistic and not reflect the true knowledge about the influence of maternal depression on child attachment style. The quality assessment was conducted by two reviewers independently (A.Ś., K.K.). The inter-rater agreement was calculated (Kappa = 0.743) and all disagreements in assessment were resolved by the third author (EB) (see Appendix A). As blind assessment has been found to have little influence on this procedure [38], the authors of the papers were not hidden during the quality evaluation process. The following criteria were used to assess the quality of the presented studies: (a)Sample size: an adequate minimum sample size was regarded as 30 participants in each compared group;(b)Representative of study population: representative and adequate study sample (should vary in demographic characteristics);(c)Low attrition: below 30% of the initial sample;(d)Validated measuring methods: standardized with given validity and reliability; proper adaptation if the measurement method was conducted in a different country than its origin;(e)Coding experience: the authors possessed training and experience in assessing attachment;(f)Statistical analysis: appropriately described and correct;(g)Assessment procedure: double-blind procedure in the attachment assessment;(h)Conclusions which were consistent with results;(i)Aim of the study: direct/indirect assessment of the influence of depression on attachment style.

The quality of the research was evaluated using a points system (+1 = yes, 0 = unclear, −1 = no) for all specified criteria. The maximum amount of points that a study could achieve was 9 and the minimum was 0: a possible evaluation below zero would be awarded a value of 0. The quality score of every study is given in last column of Table 1 and Table 2.

## 3. Results

The information about the papers, their design and research questions, sample size, methodology, main findings, and quality score were placed in two tables. Table 1 includes studies in which a clinical interview was used to diagnose depression and Table 2 for self-report questionnaires. The majority of the articles were published at some point during the last 10 years, i.e., 2007 to 2018 (*n* = 15; 52%), nine articles were published between 2000 and 2007 (31%) and only a few were published before 2000 (*n* = 5; 17%). Except for five studies which had a cross-sectional design [39,40,41,42,43] most studies were longitudinal. A detailed description of demographic data can be found in Appendix A.

### 3.1. The Influence of Major Depression on Attachment Style

Four studies evaluated the relation between maternal depression in the lifespan and child attachment style [25,44,45,46], two of which showed no such association [45,46]. Carter et al. [25] reports a higher incidence of insecure attachment among infants of a group characterised by depression with comorbidities compared to those of a ‘pure’ depression group and a no-symptoms group. A study of infant attachment and prenatal depression in mothers with a history of major depressive episodes [44] found the effect of such depression to be moderated by less than optimal maternal parenting quality. A history of major depressive disorders was found to affect child attachment style only when depression coexists with another mental disorder, or when it appears prior to the antenatal period. 

### 3.2. The Influence of Prenatal Depression on Attachment Style

Three empirical studies have evaluated the associations between maternal prenatal depression and child attachment patterns [44,46,63]. Only one of these was conducted on a varied and representative sample with socio-economic diversity; it found no relationship between insecure and disorganised attachment patterns and maternal prenatal diagnosis of depression, regardless of its severity, or maternal depressive symptoms during pregnancy [46]. In addition, no association was found between child attachment style and the maternal experience of loss and prenatal depression, even in studies involving women after stillbirth [63]. Nevertheless, infants born subsequent to a previous stillbirth were significantly more likely to express disorganised attachment patterns than those in the control group. Only one study found a significant association between maternal prenatal depression and a disorganised attachment style by the child [44]; this was observed in mothers with a history of major depressive episodes.

### 3.3. The Influence of Postnatal Depression on Attachment Style

The relationship between the occurrence of maternal depression symptoms in the postpartum period and child attachment style was evaluated in 28 of the studies included in this review. Of these, 13 found no such relationship [39,41,42,43,44,46,56,57,58,61,62,63,64], four studies found a weak influence [35,49,54,60] and 11 found a moderate association [40,45,47,49,50,51,52,54,58,64]. The studies linked the effects of postnatal depression to a variety of factors. The most important of them are socio-economic status, type and timeframe of depression measurement. 

#### 3.3.1. High-Risk Groups

The high-risk groups included participants with psychosocial and/or socio-demographic risk factors, i.e., poor financial situation, adolescent mothers, mothers with a history of stillbirth, mothers of children with very low birth weight. Six papers found no relation between maternal depression and child attachment style [39,43,44,55,56,62], three showed a weak link [35,53,59]. Two studies found a significant interaction [45,50]; however, both used a structured interview to measure depression. In contrast, out of six studies that found no dependence, only one was based on structured interviews [44].

Maternal postpartum depression in high-risk groups was not found to be a significant predictor of attachment security/insecurity among mothers with borderline personality pathology [39], mothers with histories of major depressive episodes [44], or women who have experienced stillbirth just before the current pregnancy [63].

A study by Lyons-Ruth et al. [35] found that high-risk infants from families demonstrating the combined effects of poverty, maternal depression and caretaking inadequacy had a significantly higher rate of insecure-disorganised attachment than a community sample. Measelle and Ablow [54] found higher maternal depressive symptoms to be associated with higher levels of pro-inflammatory cytokines, but only among children with insecure attachment style. In the study with adolescent mothers [60], depression was a marginal link with an attachment style. The most important factor in their case was not depression, but maternal sensitivity.

#### 3.3.2. Type of Measurement and Timeframe of the Examination of Depression

In 12 studies, depression was measured using psychiatric or structured interviews. In nine of them, depression had a significant impact on the child’s attachment style [25,40,45,47,48,49,50,51,52]. Only in three studies was no significant correlation observed between the diagnosed depression and the child’s insecure attachment style [41,44,46]. Two of the three studies had very small sample size [41,44]. In contrast, 17 studies measured the depressive mood using self-report inventories; of these, only four found it to have a significant relationship with the child’s attachment style [53,55,59,65].

The studies varied with regard to the time when maternal depression was measured. Generally, although correlations were observed between attachment style and maternal depression measured while the child was young, i.e., up to three months, these relationships became less significant when maternal depression was measured at later stages. Interestingly, studies that evaluated depression only at later stages, i.e. between six and 18 months, found no such relationship [35,39,41,43,58]. This is best illustrated by Bigelow et al. [53], who established that the risk of maternal depression decreased during the first year of life, with the sharpest decline observed between six weeks and four months, and indicated that the degree of disorganised attachment behaviour at 12 months was positively associated with the risk of maternal depression at six weeks. Similarly, Donovan and Leavitt [65] found that mothers of insecurely attached infants were more depressed than those of securely-attached children at five months; however, no such relationship was observed at 16 months. A similar pattern was observed by Righetti-Veltema et al. [51] where depressive symptomatology, as detected at three months postpartum, had an impact on attachment style 15 months later, despite the fact that most mothers no longer presented depressive symptoms at 18 months. Beeghly et al. [55] reported that maternal depressive symptoms were associated with attachment security at two to three months and at six months, and to a lesser extent at 12 months, but only amongst male infants. Such associations were no longer significant when depressive symptoms were measured at 18 months. Tomlinson et al. [50] reported the same association at two months postpartum. 

No diminished relationship was observed between maternal depression and attachment style in later months in Gravener et al. [40] and Toth et al. [48]. The two studies found that the distribution of attachment classifications measured at 20 months differed significantly between a group of infants and mothers with postnatal major depression and a control group of mothers with no depressive symptoms, with lower rates of secure attachment and higher rates of disorganised attachment being found in the depressed group. A similar relation was also found at four, 12 and 15 months by McMahon et al. [49].

Measelle and Ablow [54] report a very weak association between depression and attachment style; however, in this case, depression was measured according to the mean of two measurements taken at the fifth and seventeenth month postpartum by self-report inventory. Other studies report no association between attachment style and depression measured at six weeks [63], at two months [62,63] and at six months [44]. 

#### 3.3.3. Studies Conducted on Representative and Large Groups

Ten out of the 29 papers included in this review describe studies conducted on representative community samples. Five of them report a significant relation between maternal depression and child attachment style [48,49,52,53,59], while the other five did not [46,58,61,63,64].

Three of the studies were conducted on very large samples. Of these, Tharner et al. [46] (*N* = 627) and Sagi et al. [61] (*N* = 758) found no relationship between the maternal postpartum depression and child attachment style. However, Zaslow et al. [59] (*N* = 7894) found attachment insecurity to be positively associated with maternal postpartum depression, although it is important to note that maternal depression was measured only at nine months postpartum and that depression was also linked to poor parenting practices and household food insecurity.

### 3.4. Factors that Modify Depression—Attachment Link 

#### 3.4.1. Demographic Factors

Of the six studies with a focus on maternal age [42,46,49,53,55,56] none found it to have any connection with attachment style. Even when maternal age correlated with depression risk, the indirect effect was also not significant [53]. The associations between maternal education and attachment security are unclear: while three studies report a lack of association [46,59,60], one of them examining only young mothers aged 19 years or under [60], two other studies [49,64] indicated a strong relationship between the level of maternal education and child attachment. Interestingly, Clark-Stewart et al. [64] reported that maternal education was more important for attachment security than family structure (single vs. dual parent families), and McMahon et al. [49] indicated that a lower level of education was connected with more severe depression, as well as with an insecure internal working model and hence with child insecurity; however, the presence of unequal or small sample sizes in both studies may limit these findings.

Four studies analysed family income or socioeconomic status (SES) individually or as a cumulative demographic risk, but with inconclusive results. Beeghly et al. [55] revealed a weak correlation between child perceived security and familial demographic risk factors, including SES, family income, child sex, maternal age, education and marital status; however, this was only observed for boys. Clarke-Stewart et al. [64] examined the association between child outcomes and family income to need, and found it to have a strong correlation with attachment security. De Falco et al. [56] examine lower child attachment security in cases characterised by a co-occurrence of psychosocial risk factors, such as low family SES and maternal psychopathology, with socio-demographic risk factors, such as young age and single parenting, and compared it with cases demonstrating socio-demographic risk alone. No significant direct correlation was observed between family SES and attachment. Tharner et al. [46] report that although the monthly family income was the lowest in the resistant attachment group, no significant income differences existed between any of the attachment groups; even so, it should be noted that the study participants had a relatively high socioeconomic status.

#### 3.4.2. Psychological Factors

Two studies found no such association between maternal psychological functioning and attachment security [46,56]. Of these, De Falco et al. [56] highlighted the small variability of their sample, which included only mothers with socio-demographic or psychosocial risk factors.

Bigelow et al. [53] analysed the association between attachment style and mother’s appropriate vs. non-attuned mind-mindedness. A significant negative correlation was only found between the degree of disorganized attachment behaviour by the child, measured at 12 months, and maternal appropriate mind-mindedness, measured at four months. However, an analysis of the mediating role of appropriate mind-mindedness showed no significant association between maternal depression at six weeks postpartum and the degree of disorganized attachment behaviour by the child measured at one year.

Donovan and Leavitt [65] assessed the relationship between attachment security and maternal attributional style, as well as the perception of control over infant crying, i.e., a low, middle or high illusion of control, by the mother. No association was found between attributional style and attachment status, but high illusion of control by the mother was associated with insecure attachment. However, it should be mentioned that mothers with high illusion of control were also the most depressed.

Bergman et al. [58] found a significant relationship between maternal pre- or postnatal anxiety and attachment security: securely-attached children tended to have mothers with lower levels of postnatal anxiety.

#### 3.4.3. Gynaecological-Obstetrics Factors

Gestational age at birth and APGAR score [46], birth weight and caesarean section, the use of reproductive technology, number of days spent at hospital after delivery and singleton/multiple pregnancy [57] have been found to be associated with attachment security. Two studies examined also the role of unplanned pregnancy [50,57], but even though an unplanned pregnancy was related to early maternal depression [50], none of the studies found it to influence attachment security. Mehler et al. [57] noted that although securely-attached preterm infants tended to be breastfed more frequently than those who were insecurely attached, the difference was not significant.

Furthermore, higher rates of insecure attachment [57] or avoidant attachment [46] were observed in firstborns. Children seen by their mothers soon after birth, i.e., between 30 minutes and three hours, were also more securely attached [57]. Hughes et al. [63] revealed that infants born subsequent to stillbirth were more likely to express disorganized attachment patterns than those born after uncomplicated pregnancy. These differences were strongly predicted by maternal unresolved mourning. Even though holding the dead infant and performing a funeral was not significantly associated with disorganized attachment of the next child, its likelihood was higher when mothers had seen the dead infant. 

#### 3.4.4. Infant Factors

Our findings reveal an equivocal association between infant sex and attachment security among the studies. Five studies showed no such association [44,46,49,57,61], while four found otherwise [42,45,51,55]. The latter four clearly indicate that boys may be more vulnerable to early caregiving risks such as maternal depression, with negative consequences for mother-child attachment security. 

Infant behaviour problems were significantly associated with attachment style [52]. Predictably, the mothers and fathers of secure children reported fewer behavioural problems in their infants. However, a more detailed analysis of the studies included in the current review revealed that parenting difficulties, poorer social support and maternal stress were better predictors of attachment style. A lack of resources available for parenting resulted in poor boundary-setting for children, which contributed to insecure attachment and behavioural problems. Moreover, these factors were not influenced by maternal depression. Neither infant development status [55] nor infant temperament [61,65] predicted attachment style. 

#### 3.4.5. Social Support and Single Parenting

Further equivocal results were associated with single parenting. Three studies showed marital status to have no influence on attachment style [42,46,56] while one study found children in one-parent families to be less securely attached to their mothers [64]. It is important to note that the single women in the study reported more symptoms of depression and were poorer than mothers in two-parent families. 

The results related to partner support are not straightforward. While Donovan and Leavitt [65] indicated no influence on attachment style, other studies, including Tomlinson et al. [50], present contradictory results. The most surprising findings were provided by Tarabulsy et al. [60], who showed that greater levels of support from a partner was associated with insecure attachment. As this study included only adolescent mothers, the authors provided three hypotheses about this result: firstly, that an adolescent father may cause problematic father-infant interactions; secondly, that a father may be more disposed to being helpful when he sees that the mother of his child has emotional or maternal difficulties; thirdly, that the mother may be afraid of losing the father’s affection and may not be as sensitive to her infant as she would be under normal circumstances [60]. 

Regarding the influence of marital satisfaction on attachment security, some studies indicate that when mother was satisfied with her relationship, there was a greater chance that she would have a securely attached child [45,61]; however, Cicchetti et al., [52] found the opposite. Two studies viewing the issue of support in a much broader context, including family and friends [52,55] found that maternal social support does not attenuate the negative association between maternal depressive symptom trajectories and child attachment security. However, Donovan and Leavitt [65] report that mothers with alternative infant care to cover non-work related activities tended to have more securely attached children. 

#### 3.4.6. Parenting and Emotional Availability

Two studies explored the relations between the internal working model of relationships and maternal depression, and infant attachment style [49,60]. Both showed that autonomous mothers tended to have securely attached infants. In addition, McMahon et al. [49] indicated that maternal working models of early loving relationships buffered the effects of depressive symptoms on the development of an insecure attachment style. However, Tarabulsy et al. [60] indicated that the association was more complex, and highlighted that maternal sensitivity played a role. These maternal working models were no longer a significant predictor of infant attachment security when maternal sensitivity, partner support and maternal depression were considered together. These findings are consistent with studies that took the emotional availability of the mother into account [25,61], all of which found greater emotional availability to be associated with a greater chance of the child developing a secure attachment style. In addition, Gratz et al. [39] indicated that maternal emotional dysfunction influenced the insecure attachment style, but did not determine whether it could be considered as a moderator for the depression-attachment relationship. 

Finally, the influence of parenting quality was evaluated in four studies [40,44,51,59] which showed that parenting quality, time devoted to play and interaction with the child affected the attachment style. Zaslow et al. [59] indicated that positive parenting mediated the association between maternal depression and insecure attachment style. Depression was negatively associated with positive parenting, which in turn was negatively associated with insecure attachment style. In contrast, Gravener et al. [40] proposed that maternal self-criticism may have a significant influence on attachment style.

## 4. Discussion

The literature search revealed 29 papers describing empirical studies of the effect of maternal depression on infant attachment in the first two years of a child’s life, more specifically, the role of a lifetime history of major depression or of depression that occurred during the pre- and/or postnatal period. These studies, therefore, examined mothers from a range of backgrounds, and with considerable variation in symptom severity, duration and dynamics; however, despite such diversity, most studies do not indicate that maternal depression plays any role in determining infant attachment. Nevertheless, 13 of the 29 studies included in our present analysis indicated no such relationship. Therefore, the present review attempted to identify the factors behind such divergent results. 

The distinction between major depression and depressed mood turned out to be crucial. If depression was diagnosed by structured interview, it was commonly found to have a significant influence on child attachment style, as demonstrated in nine studies out of 12. However, if questionnaires such as the CES-D or BDI were used, depressive mood was commonly found to be unrelated to attachment style, as noted in 12 studies out of 17. Hence, the use of an accurate and reliable diagnosis of depression is crucial in a study, and should be one of the gold standards for studies on the effects of depression on attachment. 

Our findings do not confirm that antenatal or prenatal depression plays any role in infant attachment. Only one examined study identified an association between antenatal depression and attachment style, but all the women participating in that study had a history of major depression [44]. The presence of antenatal or prenatal depressive symptoms increases the risk of occurrence of an insecure attachment style only when factors such as low parental quality [44] or comorbid diagnosis [25] coexist with depression. 

Our investigations showed that even in case of high-risk groups [35,39,43,54,56,57] depression did not affect the attachment style, or that a very weak association existed [60].

The studies that do take into account postpartum depression, however, suggest that it does have an effect on attachment, but only under certain conditions, such as the time at which depression was diagnosed. It is worth stressing, however, that the examination of postpartum depression should be made before the sixth month after birth. Most studies that included a diagnosis of depression after this time showed no relationship between depression and attachment style. 

It is likely that the differences between results may be attributed to the fact that the participants of some studies had a long history of depressive symptoms and suffered from postpartum depression, while in others, major depression had only begun after the recent delivery. If only severe and chronic symptoms of depression increase the risk of insecure attachment, the question is whether these relate to the effect of depression or the effect of factors that usually accompany mood disorders (e.g., parental quality, maternal sensitivity, emotional availability). 

However, our findings confirm the claim that depressive mothers are not a homogeneous group. This is in line with the view that depressive mothers are able to create very good conditions for raising their children, regardless of the severity of symptoms [66], and that no significant relation exists between depressive symptoms and quality of infant care [67]. Therefore, the role of maternal sensitivity, emotional availability and time devoted to a child is more important than the presence of depression itself [25,39,60,61]. The need for care enforces an increase in activity and focuses the mother’s attention on the child’s needs, which can reduce the severity of depression symptoms. The difficulties she experiences can also be interpreted as related to the new role. If so, the possible negative effects of depression might be compensated by maternal involvement in childcare.

The most surprising results of our systematic review are associated with breastfeeding. Among 29 studies, only two took this factor into account [42,57]. It is unclear why this factor is overlooked by so many studies, despite the fact that a body of data suggests that longer duration of breastfeeding is associated with attachment security [46,68,69], and only a few studies show no such association [70]. Verifying whether breastfeeding moderates the impact of depression on attachment appears a very interesting topic, especially as shorter breastfeeding duration is associated with postpartum depression in almost all studies [71]. Thus, future studies should consider breastfeeding and its duration to verify its role as a factor that, when combined with maternal depression, affects attachment security.

In summary, our findings indicate that maternal depression is found to be linked to attachment security more often when it occurs postnatally and when is diagnosed by structured interview [25,44,45,47,48,49,50,51,52] and not by self-reporting measures. The findings also indicate factors that increase the likelihood of maternal depression becoming a risk for attachment insecurity. Such identified factors include severe and long-lasting depressive symptoms, coexistence of depression in the pre- and postnatal period and the family being in a difficult socioeconomic situation. The longitudinal studies included in the review also indicate that maternal depression changes with time [51], and as such, its effect might depend on the time of measurement (up to six months postpartum). This effect has also been found to vary between male and female infants [55]. 

Our findings have applications for clinical practice. Firstly, they indicate that if maternal depression is to be diagnosed, it should rather be done by clinical interview and not self-reported questionnaire, as the latter is susceptible to social desirability effects. In societies with high respect and high expectations related to the maternal role, women might feel reluctant to reveal a low mood or other depressive symptoms that may be inconsistent with socially expected maternal joy and satisfaction. A standardized interview conducted by a well-trained and supportive clinician provides more accurate data and creates better circumstances for opening a discussion of the different experiences of a new mother. In addition, while a clinical interview allows postnatal depression to be diagnosed even after a few weeks, questionnaire methods only indicate the presence of dispiritedness and worsened mood in the mother. Secondly, new mothers should be screened for depression reasonably early on, as it is the incidence of depression in the first six months postpartum that affects a child’s attachment security. Early detection of depression also allows for earlier implementation of proper intervention. 

### Strengths and Limitations of the Study

The present review has both strengths and limitations. Most importantly, it differs from other literature reviews in this area by directly analysing the relationship between maternal depression and child attachment [29,30]. Similarly, none of the four meta-analyses conducted so far have examined the distinction between antenatal/prenatal and postnatal depression [72,73,74,75]. Only Martins and Gaffan [74] stated the age of the child at which the symptoms of depression in mothers were measured as part of the studies covered in their meta-analysis.

Another strength of our study is that it performs a rigorous quality assessment of each included manuscript. The assessment was performed in compliance with Cochrane’s recommendations [37]. Therefore, the final conclusions could be formulated while taking into account the potential risk of bias resulting from factors such as non-representative and inadequate sample size [56].

A further strength, but also a limitation, is that the analysed studies were diversional: they have a longitudinal design [53,62,64], use reliable measures to identify depression and classify attachment [44,63], and are based on large representative samples [46,59]. 

On the positive side, this diversity of source materials highlights the role played by the time at which depressive symptoms were measured or diagnosed in determining their impact on attachment. On the other hand, some studies were based on smaller samples [65] or included participants with particular characteristics, such as adolescent mothers [60] or women with a history of stillbirth [63]. Although such unique features can help better understand the situation of these particular groups of women, it might not be possible to generalize these results. As the strange situation procedure has frequently been used to classify infant attachment, such studies return comparable results, and although studies may refer to insecure attachment style, they conceptualize it in the same way. Most of the included papers are based on studies with moderate to high quality (Table 1 and Table 2), indicating that their findings are reliable, irrespective of whether they confirm any depression-attachment link. 

Another limitation is that some studies were based on cross-sectional designs [39,40,41,42,43] and hence cannot be used to indicate predictive risk. Therefore, although none of them indicate the existence of such a relationship, it is impossible to conclude that it does not exist.

In addition, as one of the inclusion criteria was English as the language of the publication, further data may exist in papers published in other languages. Furthermore, the size of the samples employed by the studies varied, and only a few included large, representative groups; the participants varied with regard to demographic characteristics, and family and socioeconomic status, and although the studies were conducted in several countries, the vast majority of participants were White-Caucasian women, which might limit the generalization of the findings to other ethnic groups. Finally, as the review included studies with a focus on the association between maternal depression and attachment security, its findings cannot be extended to other effects of maternal depression, such as infant cognitive development or behavioural problems; nevertheless, some of the studies included addressed also these broader issues [64]. 

Another important consideration is that there is little, if any, replicability between individual studies. Some of the reviewed studies confirm the impact of depression, while others do not, with only some of the discrepancies being explained by factors such as the time at which depression was measured [53,58]. Nevertheless studies conducted under similar conditions and with similar samples differ in their results [39,45,49,64]. Such divergence is clearly visible in the light of recent reports from the ManyLabs project, which fails to replicate many psychological studies [76]. Despite so many years of research in this area, there is still a need for replication. 

## 5. Conclusions

Our findings indicate that the association between maternal depression and infant attachment is a complex and dynamic one. It is affected by such factors as the type of depression, method and time of measurement, as well as a number of maternal factors, such as education or self-criticism, and also by infant factors, such as being male or firstborn. Therefore, future studies in this area should consider possible confounding factors, and a multifactorial and prospective approach should be applied when designing their research protocols. Further studies should also attempt to incorporate greater cultural and ethnic diversity and aim to confirm, or refute, the assumption that the effect of maternal depression on infant attachment security is universal under similar circumstances. 

## Figures and Tables

**Figure 1 ijerph-17-02675-f001:**
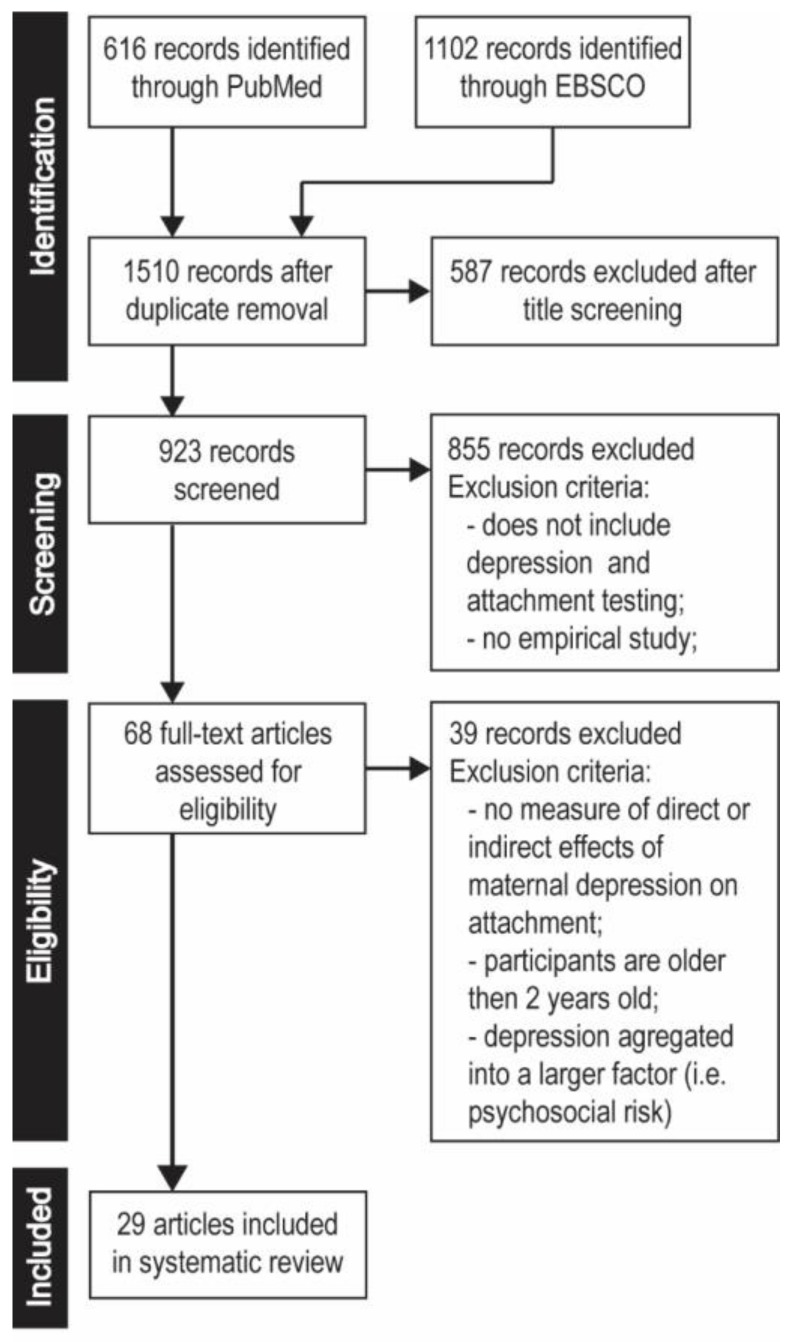
Flowchart of the search and systematic review process.

**Table 1 ijerph-17-02675-t001:** Details of the studies measuring the impact of maternal depression on attachment security which used the clinical interview to diagnose depression.

Study (Year), CountryStudy Design.	Main Objectives/Research Questions	Participant Details/Sample SizeN; M Mother’s Age (Standard Deviation, SD)	Attachment and Depression Measurement: Name of a Tool and the Time When Administered (in Trimester if Prenatally or Months if Postnatally)	Results	Study Quality Score
[47] (2016), Denmarklongitudinal study	To examine the role of personality disorders in the association between maternal postpartum depression and infant–mother attachment in a low-risk sample.	Mothers with personality disorders (PD) and mothers from low-risk sample. *N* = 80; (M age = 30.5, SD = 4.12)	SSP: 13 months;PSE: 2 months;EPDS: 2 months	Postpartum depression was associated with attachment insecurity only if the mother also had a personality disorder diagnosis. Infants of depressed mothers without co-morbid personality disorders did not differ from infants of mothers with no psychopathology. These results suggest that co-existing personality disorders may be crucial in understanding how postpartum depression impacts on parenting and infant social-emotional development..	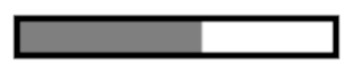 5Low attrition rate, validated methods, clear statistical analysis and direct measurement of depression-attachment influence, double-blind procedure used by experienced coders. Inadequate sample size and no representative group of mothers from low-risk sample.
[44] (2013), USAlongitudinal study	To examine associations between maternal antenatal depression and infant disorganization at 12 months. A secondary aim was to test the roles of maternal postpartum depression and maternal parenting quality as potential moderators of this association.	Only mothers with at least one major depressive episode before pregnancy.N = 79;(M *age* = 30.3; SD = 5.4)	SSP: 12 months;BDI-II: 1st and 2nd trimester; 6 and 12 months;SCID-I/P: 1st trimester	Attachment disorganization was strongly correlated with antenatal depression. Maternal parenting quality moderated this association, as exposure to higher levels of maternal depressive symptoms during pregnancy was only associated with higher rates of infant disorganized attachment when maternal parenting at three months was less optimal. No significant effects were found for postpartum depression.	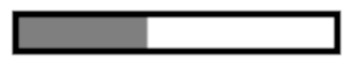 4Adequate sample size, validated methods, clear methodology and statistical analysis, direct measurement of depression-attachment influence. No information about attrition rate, double-blind procedure or coder experience, no representative high-risk study population.
[46] (2012), Netherlandslongitudinal study	To examine the effects of maternal history of depressive disorder and the effects of depressive symptoms during pregnancy and early postpartum period on attachment insecurity.	A varied and representative sample with socio-economic diversity.*N* = 627; (M *age* = 32; SD = NA)	SSP: 14 months;BSI: 2nd trimester and 2 months;EPDS: 2months; CIDI: 3rd trimester	Insecure and disorganized attachment patterns were not related to maternal lifetime diagnosis of depression regardless of its severity. Higher maternal BSI depression score during pregnancy and BSI/EPDS score during postpartum period were not related to infant-mother attachment insecurity or disorganization at 14 months.	 9Adequate and representative sample size, low attrition rate, validated methods, clear statistical analysis and direct measurement of depression-attachment influence, double-blind procedure used by experienced coders.
[40] (2011), USAprospective cross sectional	To examine associations between maternal depression and maternal expressed emotion (self-criticism, child-criticism), child internalizing and externalizing behaviors, and attachment insecurity.	Mothers with major depression with at least high-school education and good or very good socio-economic status.*N* = 198; (M age = 31.7; SD = 4.68)	SSP: 20 months; BDI: 20 months;DIS-III-R: 20 months	The depressed and nondepressed groups differed significantly regarding the main study variables, with depressed mothers evidencing higher child-criticism and self-criticism and having toddlers with higher levels of internalizing symptoms, externalizing symptoms, and attachment insecurity. Results revealed that children of mothers with higher self-criticism had a significantly higher probability of being classified as insecurely attached. However, child-criticism was not a significant mediator of the association between maternal depression and child attachment insecurity.	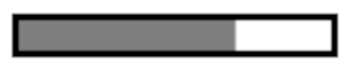 8Adequate sample size, validated methods, clear statistical analysis and direct measurement of depression-attachment influence, double-blind procedure used by experienced coders. No information about attrition rate, no representative low-risk study population.
[48] (2009), USAlongitudinal study	To examine the impact of maternal depression on attachment security and on the representation of the parents by the child.	Mothers with major depression with at least high-school education and good or very good socio-economic status.*N* = 131;(M age = NA; SD = NA)	SSP: 20 and 36 months;BDI: 20, 36 and 48 months;DIS-III-R: 20 months	At 20 and 36 months, the distribution of attachment classifications differed significantly between the depressed and nondepressed groups. Lower rates of secure and higher rates of disorganized attachment were found for the depressed group.	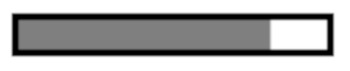 7Adequate and representative sample size, validated methods, clear statistical analysis and direct measurement of depression-attachment influence, used double-blind procedure by experienced coders. High attrition rate.
[49] (2006), Australialongitudinal study	To explore whether a mother’s own state of mind regarding attachment moderated the association between postpartum depression and insecure mother–child attachment.	A varied and representative sample with socio-economic diversity.*N* = 111;(M *age* = 31.4; SD = 4.2)	SSP: 15 months; CIDI: 4 and 12 months; CES-D: 4, 12 and 15 months	Mothers diagnosed as depressed were more likely to have an insecure state of mind regarding attachment. Infants of chronically depressed mothers were more likely to be insecurely attached. However, the relationship between maternal depression and child attachment was moderated by maternal attachment state of mind. When mothers were depressed and also had an insecure state of mind, their children were highly likely to be insecurely attached.	 9Adequate and representative sample size, low attrition rate, validated methods, clear statistical analysis and direct measurement of depression-attachment influence, double-blind procedure used by experienced coders.
[50] (2005), South Africalongitudinal study	To evaluate how early disturbances in mother-infant interactions might be related to infant attachment problems. To examine the effect of wider contextual influences on infant attachment, including the mother’s experience of depression (both at 2 and at 18 months), the degree of support she received from her partner and others, and socio-economic status.	Mothers from a high-risk (low income) group. *N* = 147;(M age = 26.8; SD = NA)	SSP: 18 months;SCID: 2 and 18 months	Mothers with postpartum depression at two months were more likely to have children who were insecurely attached. Mothers who were less sensitive, more often intrusive, coercive (maternal intrusiveness), and displayed more maternal remoteness at two months postpartum were more likely to have insecurely attached children. Mothers who were less sensitive, highly intrusive-coercive at 18 months of a child’s life also were more likely to have insecurely attached children. Mothers of secure children were more likely to report that they felt supported by their partner than insecure ones. Unwanted pregnancy, unwanted baby, maternal depression at 18 months, remote-disengaged behavior at 18 months were not related to the style of infant attachment.	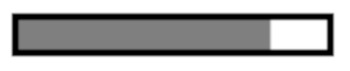 7Adequate sample size, low attrition rate, validated methods, clear statistical analysis and direct measurement of depression-attachment influence, double-blind procedure used by experienced coders. No representative group of mothers from a high-risk sample.
[51] (2003), Switzerlandlongitudinal study	To examine the effect of postpartum depression on the mother–child relationship.	Mothers with postpartum depression and control group. *N* = 70;(M age = 29; SD = NA)	SSP: 18 months;Diagnostic interview: last trimester; 3 and 19 months;EPDS: 3 and 18 months	Infants of non-depressed mothers were more likely to be securely attached to their mother. Mild or moderate depressive symptomatology, as detected at three months postpartum, had an impact 15 months later on the child’s development and on mother-child interaction, despite the fact that most mothers no longer presented depressive symptoms at 18 months. Boys were shown to be more resistant than girls.	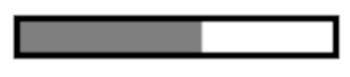 5Adequate sample size with high attrition rate. Validated methods, clear methodology and statistical analysis and direct measurement of depression-attachment influence. No information about coders experience or double-blind procedure. No clear information if study population is representative.
[25] (2001), USAlongitudinal study	To examine relations between maternal depression (in pure and comorbid forms) and mother–infant interactions, infant attachment, and toddler social-emotional problems and competencies.	A varied and representative sample with socio-economic diversity.*N* = 69; (M age = 31.9; SD = 4.88)	SSP: 14 months;SCID-NP: 4 and 14 months; CES-D: prenatally and 4 and 14 and 30 months	A history of depression and other disorders increased risk for infant insecure attachment. Higher incidence of insecure attachment was observed in infants of the comorbid group as compared with infants of the pure depression group and no-psychopathology group, but no differences between the pure depression and no-psychopathology groups were found.	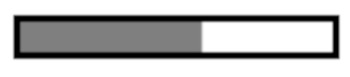 5Representative but inadequate sample size with high attrition rate. Validated methods, clear methodology and statistical analysis, direct measurement of depression-attachment influence, double-blind procedure used by experienced coders.
[52](1998), USAlongitudinal study	To examine the direct influences of maternal depression on child development, as well as the role of contextual risks that may be particularly heightened in families with depressed parents.	Mothers with at least high-school education and good or very good socio-economic status in relationships with the child’s father.*N* = 156;(M age = 31.8; SD = 4.57)	AQS: NA;BDI: NA;DIS-III-R: NA	Toddlers with depressed mothers expressed significantly more insecure attachments than did toddlers with non-disordered mothers, and this difference was not accounted for by contextual risk.	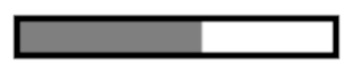 5Adequate sample size with low attrition rate, validated methods, clear methodology and statistical analysis, direct measurement of depression-attachment influence. No information about double-blind procedure or code experience, no representative low-risk study population.
[45] (1992), UKlongitudinal study	To compare the cognitive, social and emotional development of infants of mothers with main depression and/or postpartum depression with that of infants of non-depressed mothers. To assess the impact of the style of interpersonal contact associated with depression (rather than depressive symptoms) on attachment style.	Mothers of healthy borne children with major depression episode and healthy control group.*N* = 111; (M age = 28; SD = 4.3)	SSP: 18 months; EPDS: 6 and 12 months;SPI: 3 month; SADS-L: 3 and 18 months	Infants whose mothers had been depressed in the postnatal period were significantly more likely to be insecurely attached to their mothers at 18 months than infants of non-depressed mothers. No difference in outcome was found between infants whose mothers had their first episode of depression following childbirth and those who had previous as well as postpartum depression. The duration, severity of the depression and current maternal depression was unrelated to infant outcome. Women who had previous but not postpartum depression were not significantly more likely to have infants who were insecure than women with no history of depression.	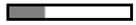 3Representative, albeit inadequate, sample size with low attrition rate, validated methods, and direct measurement of depression-attachment influence. Clear methodology but vague statistical analysis and conclusion are not consistent. No information about coder experience or double-blind procedure.

ADS-L—German version of CES-D; AQS—Attachment Behaviour Q-Sort; BDI—Beck Depression Inventory; BSI—Brief Symptom Inventory; CES-D—Center for Epidemiological Studies Depression; CIDI—Composite International Diagnostic Interview; DACL—Depression Adjective Check Lists; EPDS—Edinburgh Postnatal Depression Scale; HRSD—Hamilton Rating Scale for Depression; IDD—Inventory to Diagnose Depression; PCERA—Parent–Child Early Relational Assessment Scale; PSE—Standardized Psychiatric Interview Present State Examination; SADS-L—Schedule for Affective Disorders and Schizophrenia; SCID—Structured Clinical Interview for DSM-IV; SCID-NP—Structured Clinical Interview for DSM-IV—Non-Patient Version; SPI—Standardized Psychiatric Interview; SSP—Strange Situation Procedure; TAS-45—Toddler Attachment Sort—short version of Attachment Q Sort.

**Table 2 ijerph-17-02675-t002:** Details of the reviewed studies measuring the impact of maternal depression on attachment security; all used self-report questionnaires to diagnose depression.

Study (Year), CountryStudy Design	Main Objectives/Research Questions	Participant Details/Sample SizeN; M Mother’s Age (Standard Deviation, SD)	Attachment and Depression Measurement: Name of a Tool and the Time When Administered (in Trimester if Prenatally or Months if Postnatally)	Results	Study Quality Score
[53] (2018), USAlongitudinal study	To investigate the relationships between maternal depression risk and mind-mindedness on infant attachment behavior at one year.	A varied and representative sample with socio-economic diversity.*N* = 87;(M age = 29.4; SD = 6.5)	SSP: 12 months;CES-D: 6 weeks and 4 and 12 months	Maternal depression risk decreased over the infants’ first year, with the sharpest decline between 6 weeks and 4 months. Mothers at risk of depression when infants were 6 weeks showed less appropriate mind-mindedness at 4 months. The degree of disorganized attachment behavior by the infants at one year was positively associated with the risk of maternal depression at 6 weeks, and negatively associated with maternal appropriate mind-mindedness at 4 months.	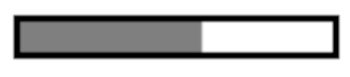 5Representative but not adequate sample size with high attrition rate, validated methods, clear methodology and statistical analysis, direct measurement of depression-attachment influence, double-blind procedure used by experienced coders.
[42] (2018), Canadaprospective cross-sectional	To examine the influence of maternal oxytocin receptor (OXTR, rs53576) genotype and cortisol secretion as moderators of the relation between maternal childhood maltreatment history and disorganized mother–infant attachment.	Mothers from low-risk sample, mainly high-educated sample. *N* = 314(M age = 32.9; SD = 4.5)	SSP: 17 months; BDI: 16 months	Marital status, employment, age, breast feeding status, self-reported ethnicity, depression, parenting stress and sensitivity, family income, number of siblings and hours per week in out of home care was not associated with attachment disorganization scores. Only infant sex was related to disorganization, insofar that males had higher disorganization scores than females. Maltreatment history more strongly predicted mother-infant attachment disorganization score and disorganized classification for mothers with more plasticity alleles of OXTR (G) and for mothers with higher SSP cortisol secretion, relative to mothers with fewer plasticity alleles and lower SSP cortisol secretion.	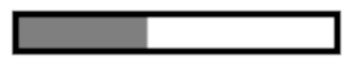 4Adequate sample size with no information about attrition rate. Validated methods, used a double-blind procedure, clear methodology and statistical analysis. No representative low-risk sample and no direct measurement of depression-attachment influence.
[54] (2018), USAlongitudinal study	To examine the association between multiple forms of early adversity—socioeconomic status disadvantage, familial stress, maternal depression, and security of attachment—and individual differences in a composite measure of pro-inflammatory cytokines and the acute phase protein CRP (C Reactive Protein).	Mothers from psychosocial and/or socio-demographic high-risk groups. *N* = 49;(M age = 24; SD = 4.7)	SSP: 17 months;CES-D: 5 and 17 months	Higher levels of depressive symptoms and insecure attachment are significantly associated with higher inflammatory load score (ILS). No significant association between maternal depressive symptoms and elevated infant ILS was observed for securely attached infants, whereas insecurely attached infants showed high levels of ILS at high levels of maternal depressive symptoms. Securely attached infants had the least amount of salivary inflammation, regardless of mothers’ levels of depression. Regions of significance analysis indicated that starting at sub-clinical levels of maternal depressive symptoms, infants classified as insecure had significantly higher ILS than all other infants.	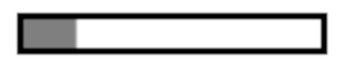 2No adequate sample size form high-risk group of mothers with no information about attrition rate. Validated methods, double-blind procedure by experienced coders, clear methodology and statistical analysis but with indirect measurement of depression-attachment influence.
[55] (2016), USAlongitudinal study	To determine whether the sex of an infant influences early vulnerability to maternal psychosocial risk, as indexed by trajectories of maternal depressive symptoms across the first 18 months’ postpartum, and toddlers’ attachment security at 18 months of age.	African American mothers from heterogeneous socio-economic backgrounds.*N* = 182;(M age = 29.5; SD = 4.37)	SSP: 18 months;CES-D: 2 and 3 and 6 and 12 and 18 months	Toddlers’ attachment security was significantly and negatively correlated with maternal depressive symptoms at 2, 3 and 6 months, marginally associated with symptoms at 12 months, but not associated with maternal depressive symptoms at 18 months. Boys’ attachment security was significantly and negatively associated with maternal depressive symptoms during the first postpartum year (at 2, 3, 6, and 12 months). Girls’ attachment security was not significantly associated with maternal depressive symptoms at any time of measurement.	 8Adequate sample size, low attrition rate, validated methods, clear statistical analysis and direct measurement of depression-attachment influence; double-blind procedure used by experienced coders. Representative African American population.
[56] (2014), Italylongitudinal study	To examine the predictors of mother–child interaction quality and child attachment security in a sample of first-time mothers with psychosocial and/or socio-demographic risk factors.	Mothers from psychosocial and/or socio-demographic high-risk groups. *N* = 40;(M age = 27.3; SD = 6.56)	AQS: 18 months;SCL-90-R: 3 months;EPDS: 3 and 6 months	Attachment security was not significantly associated with young maternal age, single parenting, psychopathological symptoms or low family socio-economic status. When groups were divided according to risk factors (socio-demographic only, psychosocial only, both factors) only socio-demographic factors were significantly associated with attachment security. Postpartum depression showed no association with either mother–child emotional availability or child attachment security.	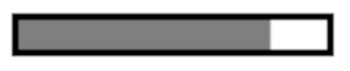 1No representative and no adequate sample size, with no direct measurement of depression-attachment influence. Unclear methodology and statistical analysis. Validated methods, SSP procedure performed by experienced coders but no information about double-blind procedure. Low attrition rate.
[39] (2014), USAprospective cross-sectional study	To examine the extent to which maternal borderline personality pathology and related emotional dysfunction (including emotion regulation difficulties and emotional intensity/reactivity) are connected with infant emotional regulation difficulties. To examine the moderating role of mother–infant attachment in the relations between maternal borderline personality pathology and related emotional dysfunction and infant emotional regulation.	Mothers with borderline personality disorder. *N* = 101; (M age = 28.7; SD = 5.6)	SSP: between 12 and 23 months;DASS: 12 months	High scores in the depression subscale of DASS correlated positively with a high symptomatology of the borderline personality (BP). No link was found between depressive symptoms and attachment classification. Results suggest that it may be the emotional dysfunction associated with BP, rather than the presence of clinically relevant BP pathology per se, that places infants of mothers with BP pathology at risk of negative outcomes.	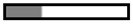 3No adequate sample size for high-risk group of mothers with low attrition rate. Validated methods, double-blind procedure used by experienced coders, clear methodology and statistical analysis but indirect measurement of depression-attachment influence.
[57] (2011), Germanylongitudinal study	To examine the effect of the time of first meeting between mother and very low birth weight (VLBW) newborn on the establishment of a secure attachment behavior; to indicate the role of maternal depression, social support and pregnancy history.	Mothers of very low birth weight (VLBW) newborns.*N* = 52;(M age = 31.2; SD = 5.1)	SSP: 12-18 months c.a. (corrected age);ADS-L: 0 and 3 and 12 months c.a.	Not seeing child within 30 min to 3 h after birth and first born child were identified as the best predictors for insecure attachment behavior. Maternal factors including age of the mother, degree of depression at three time points, social support, social status of mother and father, and pregnancy history did not significantly differ between children with secure and insecure attachment.	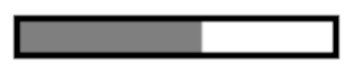 5No representative or adequate sample size with good attrition rate. Validated methods, used double-blind procedure, clear methodology and statistical analysis, direct measurement of depression-attachment influence. No information about SSP coder experience.
[58] (2010), UKlongitudinal study	To determine whether prenatal cortisol exposure predicts infant cognitive development and to evaluate how infant–parent relationship moderates this effect.	A varied and representative sample with socio-economic diversity.*N* = 125;(M age =36.6; SD=4.1)	SSP: 17 months; EPDS: 17 months	Cortisol level during pregnancy and postnatal depression were not associated with infant attachment style. Securely attached children had mothers who reported lower levels of postnatal state anxiety.	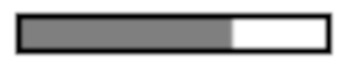 5Adequate and representative sample size, validated methods, clear statistical analysis, double-blind procedure used by experienced coders. High attrition rate and indirect measurement of depression-attachment influence.
[59] (2009), USAlongitudinal study	To examine the associations between household food security (access to sufficient, safe, and nutritious food) during infancy and attachment and mental proficiency in toddlerhood.	A varied and representative sample with socio-economic diversity.*N* = 7894;(M age =27.3; SD=13.1)	TAS-45: 24 months;CES-D: 9 months	Food insecurity has no significant direct association with being insecurely attached. Instead, food insecurity works indirectly through depression and parenting practices to influence insecure attachment. Attachment insecurity is positively associated with depression.	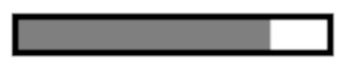 7Adequate and representative sample size, low attrition rate, validated methods, clear statistical analysis and direct measurement of depression-attachment influence. No information about double-blind procedure and coders experience.
[60] (2005), Canadalongitudinal study	To replicate the finding that a significant association exists between maternal state of mind, maternal sensitivity, and infant attachment security, but also that maternal sensitivity only partially mediates the association between maternal state of mind and infant attachment. The second objective was to consider whether paths to infant attachment security may exist that do not originate with maternal state of mind.	Adolescent mothers up to 19 years old. *N* = 64;(M age = 17.4; SD = 1.5)	AQS: 15 and 18 months; CES-D: 6 months	Significant (but weak) associations were documented between maternal attachment state of mind and maternal sensitivity, as well as between maternal sensitivity and infant security. Association between maternal sensitivity and attachment security remained significant even when other variables were statistically controlled. A marginal link was observed between maternal depression and infant attachment security. Furthermore, there was no association between depression and maternal sensitivity.	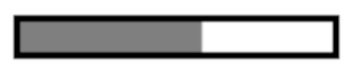 5Adequate sample size with no information about attrition rate. Validated methods used by experienced coders, clear methodology and statistical analysis and direct measurement of depression-attachment influence. No information about double-blind procedure and no representative group of adolescent mothers.
[61] (2002), Israellongitudinal study	To predict attachment classifications from: (a) relevant control variables (socio-economic status, marital relation, knowledge of infant development), (b) a maternal variable (sensitivity or depression) or child variable (gender or temperament), (c) a childcare variable (type of care at 12 months, amount of nonmaternal care, age of entry into nonmaternal care, stability of care, different types of care, and infant–adult ratio), and (d) the interaction between the two selected (mother/child and child-care) variables.	Jewish mothers of healthy children with socio-economic diversity.*N* = 758;(M age = 29.2; SD = 4.78)	SSP: 12 months;DACL: 6 and 12 months;BDI: 6 months	Center-care adversely increased the likelihood of infants developing insecure attachment to their mothers as compared to infants who were either in maternal care, individual non-parental care with a relative, individual nonparental care with a paid caregiver, or family day-care. Mothers found to be more sensitive were more likely to have securely attached infants regardless of care type (maternal, family, nanny etc.). Maternal depression, child’s gender, temperament, age (in weeks) when nonmaternal care was introduced, length of extra care (3–12 months), stability of care were not found to be significant predictors of attachment security/insecurity.	 9Adequate and representative sample size, low attrition rate, validated methods, clear statistical analysis and direct measurement of depression-attachment influence, double-blind procedure used by experienced coders.
[62] (2001), Swedenlongitudinal study	To explore the long-term impact of depressive symptomatology on mother–child interaction and on infant attachment to mothers.	No adequate information about studied sample. *N* = 45; (M age = NA, SD = NA)	PCERA: 15 and 18 months;EPDS: 2 months	Children of mothers with high EPDS scores were less curious and focused on free play situations than children of low EPDS scorers. Even if the proportion of insecurely attached children did not differ between high and low EPDS groups, those children who had mothers in the high-scoring group were more likely to show restricted levels of joy in their secure attachment behaviours.	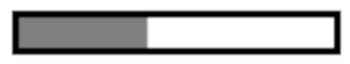 4No adequate sample size with high attrition rate. Validated methods, clear methodology and statistical analysis, direct measurement of depression-attachment influence, double-blind procedure used by experienced coders. No clear information whether study population is representative.
[63] (2001), UKlongitudinal study	To determine whether experience of loss may lead to unresolved state of mind in a mother, and whether this is associated with increasing rates of disorganization of infant attachment patterns among infants born subsequent to stillbirth. To determine whether disorganized infant attachment could be predicted by maternal symptoms of depression or anxiety, by social disadvantage, by additional experience of miscarriage or termination of pregnancy, or by whether or not the mother had seen and held her stillborn infant and had held a funeral for the infant.	Mothers who have experienced a stillbirth (after 18 month of gestation) before their current pregnancy and a control group. *N* = 106; (M *age* = 19.9; SD = 5.4)	SSP: 12 months;EPDS: 3rd trimester and 6 weeks; BDI: 26 weeks and 12 months	Infants next-born after a stillbirth were significantly more likely to have disorganized attachment style than control infants. Maternal depression was not associated with child attachment style.	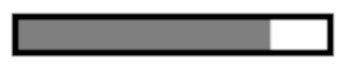 7Adequate and representative sample size, validated methods, clear statistical analysis and direct measurement of depression-attachment influence, double-blind procedure used by experienced coders. High attrition rate.
[64] (2000), USAlongitudinal study	To explore the effects of marital separation and divorce on psychological development of children in the first 3 years of life. To examine the consequences of marital separation on children’s functioning immediately after separation and to assess whether maternal background before and after marital separation may affect studied variables.	A varied and representative sample with socio-economic diversity.*N* = 340;(M age = 26.3; SD = NA)	SSP: 15 and 36 months; AQS: 24 months;CES-D: 1 and 6 and 15 and 24 and 36 months	Children in 2-parent families performed better than children in 1-parent families on assessments of cognitive and social abilities, problem behavior, attachment security, and behavior with mother. However, controlling for maternal education and family income reduced these differences, and associations with separated-intact marital status were non-significant. Hence, child psychological development was not affected by parental separation per se; it was related to maternal income, education, ethnicity, child- rearing beliefs, depressive symptoms, and behavior. Maternal depression increased behavioral problems of the child, but did not affect the security of attachment style.	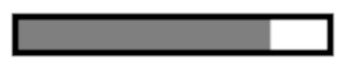 8Adequate and representative sample size, low attrition rate, validated methods, clear statistical analysis and direct measurement of depression-attachment influence, SSP procedure performed by experienced coders. No information about double-blind procedure.
[35] (1990), USAIntervention longitudinal study	To assess whether a family in high-risk circumstances (combined effects of poverty, maternal depression, and caretaking inadequacy) could benefit from family support services.	Mothers from a high-risk (low income) group and controls. *N* = 76;(M age = 24.06, SD = NA)	SSP: 12 months; CES-D: 12 and 18 months	The negative effects of social risk status were more pervasive in regard to infant attachment security than in regard to mental development, with the unsupported high-risk group as a whole differing significantly both from the supported high-risk group. Unsupported high-risk infants had a very high rate of insecure-disorganized attachment, 60%, compared to 29% for high-risk infants supported by family services and 28% for community infants.	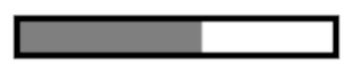 5Representative but not adequate sample size with no information about attrition rate. Validated methods, double-blind procedure used by experienced coders, clear methodology and statistical analysis, no direct measurement of depression-attachment influence.
[65] (1989), USAlongitudinal study	To examine the predictors of mother–child interaction quality and child attachment security in a sample of first-time mothers with psychosocial and/or socio-demographic risk factors.	Married mothers. *N* = 40; (M age = 27.3; SD = 6.56)	SSP: 16 and 40 months;BDI: 5 and 16 months	Insecure infant attachment at 16 months was associated with maternal perception of overcontrol, depressed mood state, and aversive conditioning to impending cries in a laboratory task at the 5-month period. Mothers of insecurely-attached infants were more depressed at 5 (but not 16) months than mothers of securely attached infants.	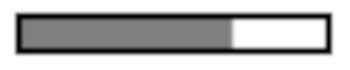 6No adequate sample size with low attrition rate. Validated methods, clear methodology and statistical analysis, direct measurement of depression-attachment influence, double-blind procedure by experienced coders used. No clear information whether study population was representative.
[43] (1986), USAprospective cross sectional	To examine whether maternal depression is a risk factor for infant development, as well as for childhood psychopathology, and whether maternal depression affect the security of attachment at twelve months of age in a low-income sample.	Mothers from high-risk (low income) group. *N* = 56;(M age = NA; SD = NA)	SSP: 12 months; CES-D: 0–9 month and 18 months	Depressed and non-depressed mothers did not differ in incidence of insecure infant attachment, nor did the maternal depression scores correlate with infant reunion behaviors in the strange situation, including infant avoidance or resistance at reunion. Mothers reporting mild to moderate depression were more likely to have securely attached infants, while mothers reporting severe depression were more likely to have infants showing unstable avoidant attachment. More surprisingly, mothers reporting the least frequent depressive symptoms were more likely to have avoidant infants.	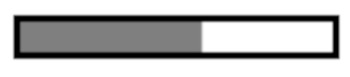 5Adequate sample size with no information about attrition rate. Validated methods, used double-blind procedure, clear methodology and statistical analysis, direct measurement of depression-attachment influence. No clear information if SSP coders have some experience. No representative high-risk population.

ADS-L—German version of CES-D; AQS—Attachment Behaviour Q-Sort; BD—Beck Depression Inventory; BSI—Brief Symptom Inventory; CES-D—Center for Epidemiological Studies Depression; CIDI—Composite International Diagnostic Interview; DACL—The Depression Adjective Check Lists; EPDS—Edinburgh Postnatal Depression Scale; HRSD—Hamilton Rating Scale for Depression; IDD—Inventory to Diagnose Depression; PCERA—The Parent-Child Early Relational Assessment Scale; PSE—Standardized Psychiatric Interview Present State Examination; SADS-L—Schedule for Affective Disorders and Schizophrenia; SCID—Structured Clinical Interview for DSM-IV; SCID-NP—Structured Clinical Interview for DSM-IV—Non-Patient Version; SPI—Standardized Psychiatric Interview; SSP—Strange Situation Procedure; TAS-45—Toddler Attachment Sort—short version of Attachment Q Sort.

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
