# Peer review of "The Effect of Maternal Depression on Infant Attachment: A Systematic Review"

_ijerph, 2020, doi:10.3390/ijerph17082675_

Round 1

Reviewer 1 Report

Comments on the manuscript

This manuscript is a systematic review of associations between maternal depression and infant attachment. This manuscript is very informative and well constructed. Basically, I do not have any major question on this manuscript. However, I have a few minor questions as below before publication.

Minor questions:

  1. Table 1: Table of the main result of the study needs caption. At my first look at this Table, I could not understand why the authors started with the study by “Carter et al. (2001), ” and I wondered why the authors did not group by study design or study quality score. Those studies may be alphabetically ordered. But, I think, those studies should be ordered by the study quality score, or by another better way.

  1. Study quality score: I could not understand exactly how to put a score to the studies. According to Table S1: Quality assessment scores, I still could not understand. Moreover, several studies which were not included in the results seem to also be shown on this Table S1?

I wonder whether this quality score is reliable, even after I read the explanation.

  1. Table S1: The authors explained about demographic variables of mothers. Table of the mothers’ demographics should be shown in the text, not in the supplementary material. But, both Tables of mothers’ demographics and study quality score are on Table S1? The study quality score should be on Table S2?

Author Response

Response to Reviewer 1 Comments

Point 1: Table 1: Table of the main result of the study needs caption. At my first look at this Table, I could not understand why the authors started with the study by “Carter et al. (2001), ” and I wondered why the authors did not group by study design or study quality score. Those studies may be alphabetically ordered. But, I think, those studies should be ordered by the study quality score, or by another better way.

Response 1: To better clarify the criteria used to group the studies presented in Table 1, we decided to split the table in two: the revised version of the manuscript now includes Table 1, presenting studies where depression was measured by interview, and Table 2, presenting studies where depression was measured by questionnaire. Furthermore, in both tables, the studies are now ordered by publication date: we agree that using alphabetical order was indeed not transparent.

Point 2: Study quality score: I could not understand exactly how to put a score to the studies. According to Table S1: Quality assessment scores, I still could not understand. Moreover, several studies which were not included in the results seem to also be shown on this Table S1? I wonder whether this quality score is reliable, even after I read the explanation.

Response 2: The procedure used to assign study quality score is described in the Method section, thus we decided not to change anything in the main text. However, we have changed Table S1 in supplementary materials. For greater transparency, the reviewers' assessments were placed in columns, and the studies in the rows of the table; thanks to this, the table is not divided into two separate parts. Below the table, we have placed a legend describing the meaning of each of the evaluation criteria. We have also added a description of our scoring system. Finally, we have improved the citations for each article in accordance with JIREPH requirements.

Point 3: Table S1: The authors explained about demographic variables of mothers. Table of the mothers’ demographics should be shown in the text, not in the supplementary material. But, both Tables of mothers’ demographics and study quality score are on Table S1? The study quality score should be on Table S2?

Response 3: The demographic characteristics of mothers has not been moved to the main text. All data in this table duplicates information from the systematic review study tables (Table 1 and Table 2). In the supplementary text on demographic data we have created a table that briefly describes the studies. This allows readers to follow the text without having to refer to the table in the main text. However, as suggested, we have changed its numbering to Table S2. We have also changed the citations in accordance with JIREPH requirements.

Reviewer 2 Report

This review was nicely conducted and written.  Good discussion of strengths and limitations.  I have three minor recommendations:

1)  Add comments to the abstract and conclusion regarding the mothers' ability to compensate for their depressive symptoms.

2) Add comments to the conclusion about future studies examining the effects of breast-feeding.

3) There are a few very minor grammatical errors throughout the paper, specifically the use of indefinite articles.

Author Response

Response to Reviewer 2 Comments

Point 1: Add comments to the abstract and conclusion regarding the mothers' ability to compensate for their depressive symptoms.

Response 1: Comments related to the abilities of mothers to compensate for depressive symptoms were added to the Abstract and the main text.

Point 2: Add comments to the conclusion about future studies examining the effects of breast-feeding.

Response 2: Remarks on the need to examine the effects of breastfeeding have been added.  

Point 3: There are a few very minor grammatical errors throughout the paper, specifically the use of indefinite articles.

Response 3: A native speaker of English has revised the manuscript to correct grammatical errors, especially those related to the use of indefinite articles, together with any typographical or lexical errors.

Reviewer 3 Report

I think this review is well written and provides a usful primer on this topic for scientists planning to conduct research in this area. 

A few suggestions:

  1. Include some dicussion of how providers/clinicians can use the information in the review to improve their practice. 
  2. Provide clearer justification for the decision to use 24 months as the cut-off age for children when excluding/including studies. I think 24 months is a fine decision, but this decision should still be explained better in the introduction.
  3. In the introduction, when providing statistics on prevelance of prenatal and postpartum depression, I suggest including the prevelance of similarly aged non-pregnant women for comparison.

Author Response

Response to Reviewer 3 Comments

Point 1: Include some dicussion of how providers/clinicians can use the information in the review to improve their practice.

Response 1: Comments on the practical value of the results have been added to the Discussion.

Point 2: Provide clearer justification for the decision to use 24 months as the cut-off age for children when excluding/including studies. I think 24 months is a fine decision, but this decision should still be explained better in the introduction.

Response 2: The decision to use 24 months as the cut-off age was added in the Introduction.

Point 3: In the introduction, when providing statistics on prevelance of prenatal and postpartum depression, I suggest including the prevelance of similarly aged non-pregnant women for comparison.

Response 3: Data to the prevalence of depression in non-pregnant women was added in the Introduction.